# Ecological Profile of the Flea Beetle Genus *Calotheca* Heyden in South Africa (Chrysomelidae, Galerucinae, Alticini)

**DOI:** 10.3390/insects15120994

**Published:** 2024-12-15

**Authors:** Maurizio Biondi, Mattia Iannella, Paola D’Alessandro

**Affiliations:** Department of Life, Health & Environmental Sciences, University of L’Aquila, Via Vetoio—Coppito, 67100 L’Aquila, Italy; maurizio.biondi@univaq.it (M.B.); paola.dalessandro@univaq.it (P.D.)

**Keywords:** Afrotropical region, *Calotheca*, ecological niche modelling, flea beetles, GIS analysis, *Searsia*, vegetation types

## Abstract

This study focuses on 25 species of flea beetles from the genus *Calotheca*, which are found in South Africa. Using ecological data like temperature, rainfall, and vegetation types, the research aims to understand what environmental factors influence where these beetles live. By using ecological niche modelling, this study identifies patterns for each species and for the entire genus. The results show that certain climate factors, such as the average temperature during the wettest and driest seasons, and rainfall in the wettest period, play an important role in the beetles’ distribution. Also, it highlights that areas most suitable for these beetles overlap with regions where their host plants, from the *Searsia* genus, are common. Our findings help us better understand the relationship between *Calotheca* species and their environment, with important implications in a biogeographic context.

## 1. Introduction

*Calotheca* Heyden 1887 is a sub-Saharan flea beetle genus (Chrysomelidae: Galerucinae: Alticini) with limited extensions into Israel and the Arabian Peninsula, particularly common in the eastern and southern parts of its range [1] (Figure 1c). Biondi et al. [2] redefined the genus and transferred to it several Afrotropical species previously attributed to *Blepharida* Chevrolat, 1836. Currently, it includes 37 described species [3,4,5,6]. Morphologically, the genus can be easily identified by the co-occurrence of the following characters: medium to large size (approximately 3.50–9.00 mm); dorsal margin of middle and hind tibiae with a distinct ciliate emargination (mce, hce: Figure 1a); sinuate and deeply impressed frontal grooves, which extend from the dorsal ocular margin to the interantennal space (fg: Figure 1b); punctate lateral striae on the pronotum, which extend from the anterior margin to the disc and are generally L- or C-shaped (pls: Figure 1b); short lateral longitudinal furrows or small dimples close to the pronotal base are sometimes visible (bf: Figure 1b); a shallow medial dimple is occasionally present (smd: Figure 1b). Based primarily on some recent investigations, *Calotheca* species are generally associated with host plants of the genus *Searsia* F.A. Barkley [=*Rhus* L. pars, cf. Moffett (2007) [7]] (Anacardiaceae) [8,9] in several vegetation types, mainly Savannah, Forest, Fynbos, and Karoo [10].

In this work, the species recorded for South Africa, including the Democratic Republic of South Africa, Lesotho, and Eswatini, are considered. Knowledge of the *Calotheca* species occurring in this area has increased significantly in the latest years. Until 2015, only 12 species were known (cf. Biondi et al. [2]), but recent studies have increased this number to 25 [3,4,5,6]. Some of these species belong to groups that have been recently revised, ensuring their taxonomy is now thoroughly assessed. Others, generally marginal to our study area, such as *Calotheca marginalis* s.l., require more in-depth studies to explain their high chromatic and dimensional variability and to better define the limits of such variability and the possible taxonomic status.

Our choice to consider only South Africa as reference area is based on the following considerations: (a) it is a well-defined area from a biogeographical point of view, where the biodiversity of the genus *Calotheca* is at its maximum; (b) this area is characterized by a well-distributed sampling effort, effort that is decidedly less in other areas of the continental Africa affected by the presence of this flea beetle genus; (c) availability of environ-mental data for this area, in particular vegetation data, which made possible many of the analyses we carried out.

Despite the great interest in the biodiversity of the Afrotropical region, the environmental needs, habitat preferences, and ecological relationships of the *Calotheca* species remain poorly understood, mainly due to the historical lack of a robust taxonomic framework.

This study aims to address these shortcomings. Starting from the species updated distribution and the topographic, temperature, and precipitation variables, as well as the vegetation types in the occurrence sites, the ecological profile of each species and the entire genus are provided, shedding light on the factors that drive their occurrence and distribution patterns. The distribution of *Calotheca* in the study area is also compared with that of the genus *Searsia*, providing a more comprehensive understanding of the ecological and biogeographical relationships between the flea beetle genus and its main host plants.

## 2. Materials and Methods

### 2.1. Study Area, Calotheca Database, and Vegetation Types

The study area encompasses the Republic of South Africa, Lesotho, and Eswatini (Figure 2a–c). We performed analyses on a dataset including 396 occurrence localities for the 25 species of the genus *Calotheca* obtained using information from field data, critical screening of the entomological literature [2,3,4,5,6], and identification of specimens preserved in the following public repositories and collections: BAQ: collection of M. Biondi, University of L’Aquila, Italy; BMNH: The Natural History Museum, formerly British Museum (Natural History), London, UK; DGF: Collection of D. G. Furth, Washington, DC, USA; MCZ: Museum of Comparative Zoology, Harvard University, Cambridge, MA, USA; NHMUK: United Kingdom, London, The Natural History Museum; MHNB: Muséum d’Histoire Naturelle de Bâle, Basel, Switzerland; MNHN: Muséum National d’Histoire Naturelle, Paris, France; MSNG: Museo Civico di Storia Naturale di Genova, Genova, Italy; MZH: Finnish Museum of Natural History, Helsinki, Finland; MZUF: Museo Zoologico dell’Università di Firenze “La Specola”, Firenze, Italy; MZLU: Museum of Zoology, Lund University, Sweden; NMNW: National Museum of Namibia, Windhoek, Namibia; NMPC: Entomologické oddělení Národního muzea, Praha-Kunratice, Czech Republic; NHRS: Naturhistoriska Riksmuseet, Stockholm, Sweden; SANC: South African National Collection of Insects, Pretoria, Republic of South Africa; USNM: United States National Museum (=National Museum of Natural History, Smithsonian Institution), Washington, DC, USA; UWCP: Poland, Wrocław, University of Wroclaw, Museum of Natural History; ZMHB: Museum für Naturkunde der Humboldt-Universität zu Berlin, Berlin, Germany; ZSM: Zoologische Sammlung des Bayerischen Staates, München, Germany. The abbreviations listed above follow the list on the website “The Insect and Spider Collections of the World” [12]. Geographical coordinates of the localities (WGS84 datum) are available from the authors upon request.

The vegetation types (Figure 3) occurring in the study area and used in our analyses refer to Mucina and Rutherford [13] and Dayaram et al. [14]. In alphabetical order, they are as follows: Albany thicket, a dense, spiny shrubland with a canopy up to 2.5 m in height and usually abundant succulents; Azonal vegetation, including vegetation types not directly influenced by the zonal biomes; Desert, an environment so dry that it supports only extremely sparse vegetation, with trees usually absent; Forest, a vegetation type dominated by tree species with overlapping canopies covering at least 75%; Fynbos, a scrubland plant community found along a narrow strip of the extreme southern coast of South Africa, composed of many species of broad-leaved evergreen shrubs; Grassland, a vegetation type dominated by grasses and other herbaceous (non-woody) plants; Indian Ocean Coastal Belt, a region of coastal dunes and coastal grassy plains in KwaZulu-Natal and Eastern Cape, from sea level to an altitude of about 600 m a.s.l.; Nama karoo, a xeric shrubland vegetation located in the interior of the western half of South Africa and extends into the southern interior of Namibia; Savannah, a vegetation type growing under hot, seasonally dry climatic conditions, characterized by an open tree canopy above a continuous tall grass understory; Succulent karoo, a vegetation type dominated by dwarf, succulent shrubs, mainly Mesembryanthemaceae and Crassulaceae.

### 2.2. Current Model Building for Calotheca and Evaluation

To estimate the current suitable areas for the *Calotheca* species in the study area, we built ecological niche models (ENMs) on all known occurrences of the genus. For this purpose, we selected as candidate predictors the 19 temperature- and precipitation-related “bioclimatic” raster variables from the Worldclim.org repository, as reported in Table 1.

To avoid potential correlation among variables (responsible of the lowering of the model’s discrimination power), we measured both the variance inflation factor (VIF), setting the threshold = 10 [15], and Pearson’s r (|r| < 0.9), following Dormann et al. [16] and Elith et al. [17], through the ‘vifstep’ and ‘vifcor’ functions of the ‘usdm’ R package [18]. The variables obtained as the analyses’ outcomes were then selected as predictors to calibrate the model.

The ecological niche modelling (ENM) was performed using the “Presence-only Prediction (Maxent)” tool in the ArcGis Spatial Analyst. This tool permits us to infer, based on a set of environmental predictors and occurrence localities (specifically, a presence-only dataset), the suitability of a certain taxon across an area, also giving marginal response curves of the predictors with respect to the predicted suitability. Its main advantage in terms of models’ discrimination capability is the possibility to calibrate and evaluate performances through a spatial jackknifing procedure [19]: the study area is divided into n random groups (n = 3, in our case) based on the Voronoi tessellation procedure [19,20,21] applied to occurrence localities; n − 1 groups are then used for calibration, while the remaining one is iteratively used for validation.

Cluster analysis was then performed using the “Clustered Heat Maps (Double Dendrograms)” tool in NCSS 2023. A heat map is a two-way display of a data matrix in which the individual cells are displayed as coloured rectangles. The bioclimatic variables are shown on the columns and the species on the rows. Each bioclimatic variable has a gradient of values reported in logarithmic scale. The order of the species was determined by the analysis itself, grouping together the most similar ones based on the variables considered. The order of the bioclimatic variables was determined similarly. The clustering method used in our analysis was the “unweighted pair group method with arithmetic mean” (UPGMA), which is very frequently applied in ecology and systematics [22].

### 2.3. Dataset and Density Analysis for Searsia Plants

The occurrence localities in the study area of *Searsia* (18208) were downloaded from GBIF (DOI:10.15468/dl.cpd7vz, 3 May 2024) with the help of the R package “coordinate-cleaner”: occurrences. Geographical coordinates of the localities (WGS84 datum) are available from the authors upon request.

The density-based clustering tool in ArcGIS Pro 3.3 was used to detect the areas where the presence points of *Searsia* are more clustered (forming spatially significant groups based on a specific clustering algorithm, see next) and the areas where they are absent or sparse. Points that are not part of a cluster are labelled as “noise”. The parameters used for this analysis are the following: “Clustering Method” = Self-Adjusting (HDBSCAN), and “Minimum Features per Cluster” = 300. Among the various clustering methods, HDBSCAN is the one most influenced by the nature of the input data. It uses variable and optimized distance intervals to separate the densest clusters from the sparsest noise. With the self-adjusting method, reachability distances are considered as nested levels of clusters. Each level of clustering results in a different set of clusters being detected. HDBSCAN chooses which level of clusters within each set of nested clusters optimally creates the most stable clusters that incorporate as many members as possible while excluding “noise”.

## 3. Results

Taking into account the VIF and Pearson’s correlation analyses, we selected a set of seven uncorrelated bioclimatic variables (BIO2: mean diurnal range, BIO3: isothermality, BIO8: mean temperature of the wettest quarter, BIO9: mean temperature of the driest quarter, BIO13: precipitation of the wettest month, BIO14: precipitation of the driest month, BIO15: precipitation seasonality), which were used both to describe the species’ climatic needs and calibrate the model.

### 3.1. Commented List of the Calotheca Species in the Study Area

Below is reported, in alphabetical order, the commented list of the 25 species of the genus *Calotheca* known for the study area. For each species, the updated geographical distribution and ecological information are reported, including those derived from our analyses:

*Calotheca carolineae* D’Alessandro, Iannella, Grobbelaar, Biondi, 2022 [11]

Distribution: Republic of South Africa (KwaZulu-Natal) [11] (Figure 2c). Endemic to the study area. Chorotype: Southern-Eastern African (SEA).

Ecology: Adults were collected at elevations between 0 and 50 m a.s.l. from November to February on plants of *Allophylus natalensis* (Sapindaceae) and *Ozoroa obovata* (Anacardiaceae) [11]. From the results of our analysis, this species appears to be closely associated with coastal forests in conditions with high humidity (Figure 4).

The three occurrence sites are characterized by the following mean values for the seven bioclimatic variables considered: BIO2 = 10.03 °C, BIO3 = 57.74%, BIO8 = 25.24 °C, BIO9 = 19.21 °C, BIO13 = 141 mm, BIO14 = 26.33 mm, and BIO15 = 53.46 mm (Figure 5).

*Calotheca danielssoni* D’Alessandro, Iannella, Grobbelaar, Biondi, 2021 [4]

Distribution: Republic of South Africa (Northern Cape, Western Cape) [4] (Figure 2c). Endemic to the study area. Chorotype: Southern-Western African (SWA).

Ecology: Adults were collected on plants of *Searsia* sp. (Anacardiaceae) at elevations between 200 and 760 m a.s.l., in April, September, October, and December [4]. Based on our analysis, this species results associated with the typical South-African vegetation, such as Fynbos, Albany thicket, and Succulent karoo (Figure 4). From a bioclimatic point of view, its eight occurrence sites are characterized by the following mean values for the seven variables considered: BIO2 = 13.75 °C, BIO3 = 55.03%, BIO8 = 13.98 °C, BIO9 = 20.84 °C, BIO13 = 48.33 mm, BIO14 = 10.78 mm, and BIO15 = 51.92 mm (Figure 5).

*Calotheca haroldi* (Baly, 1878) [23]

Distribution: Botswana, Malawi, Mozambique, Republic of South Africa (Gauteng, Limpopo, Northern Cape, North West, Western Cape), South Sudan, Tanzania, and Zimbabwe (Biondi et al., 2017). Occurrences in the study area are shown in Figure 2a. Chorotype: Eastern African (EAF).

Ecology: Adults were collected, mainly at elevations between 400 and 1600 m. a.s.l. from December to February; no information is available on its host plants [2]. From our analysis, *C. haroldi* results mainly associated with savannah environment (Figure 4). Its forty occurrence sites in the study area show the following mean values for the seven bioclimatic variables considered: BIO2 = 14.23 °C, BIO3 = 57.16%, BIO8 = 22.33 °C, BIO9 = 13.32 °C, BIO13 = 121.22 mm, BIO14 = 5.85 mm, and BIO15 = 80.41 mm (Figure 5).

*Calotheca holubi* (Jacoby, 1893) [24]

Distribution: Republic of South Africa (North West), Zambia, and Zimbabwe [2]. Occurrences in the study area are shown in Figure 2b. Chorotype: Southern–Eastern African (SEA).

Ecology: No information is available on the host plants of this species. In the study area, *C. holubi* is recorded by only one locality in North West Province (Dinokana,1294 m a.s.l.), collected in November (Biondi et al. [2]: pars). From our analysis, the occurrence site falls in the savannah environment (Figure 4) and is characterized by the following mean values for the seven bioclimatic variables considered: BIO2 = 15.31 °C, BIO3 = 55.26%, BIO8 = 23.30 °C, BIO9 = 12.23 °C, BIO13 = 109 mm, BIO14 = 1 mm, and BIO15 = 81.65 mm (Figure 5).

*Calotheca inornata* (Jacoby, 1895) [25]

Distribution: Republic of South Africa (Eastern Cape, Mpumalanga, Western Cape) [2] (Figure 2c). Chorotype: Southern-Western African (SWA).

Ecology: Adults of this endemic species were collected at elevations between 10 and 800 m a.s.l., in November, December, and January; no information is available on its host plants [2]. From our analysis, *C. inornata* results mainly associated with Fynbos and, to a lesser extent, with the savannah environment (Figure 4). The seven occurrence sites known for *C. inornata* are characterized by the following mean values for the seven bioclimatic variables considered: BIO2 = 10.28 °C, BIO3 = 54.96%, BIO8 = 15.97 °C, BIO9 = 17.94 °C, BIO13 = 105.86 mm, BIO14 = 18.57 mm, and BIO15 = 60.44 mm (Figure 5).

*Calotheca leonardii* D’Alessandro, Grobbelaar, Iannella, Biondi, 2023 [5]

Distribution: Republic of South Africa (Eastern Cape, Western Cape) [5] (Figure 2c). Endemic to the study area. Chorotype: Southern African (SAF).

Ecology: Adults were collected on plants of *Searsia* sp. (Anacardiaceae) and *Rhigozum obovatum* (Bignoniaceae) at elevations between 20 and 1100 m a.s.l., in January, March, May, July, November, and December [5]. Based on our analysis, *C. leonardii* is associated with the typical South African vegetation, mainly Albany thicket (Figure 4). The seventeen occurrence sites are characterized by the following mean values for the seven bioclimatic variables considered: BIO2 = 9.83 °C, BIO3 = 56.88%, BIO8 = 18.79 °C, BIO9 = 15.95 °C, BIO13 = 91.23 mm, BIO14 = 29.23 mm, and BIO15 = 33.80 mm (Figure 5).

*Calotheca luteomaculata* D’Alessandro, Iannella, Grobbelaar, Biondi, 2020 [3]

Distribution: Republic of South Africa (Gauteng, Limpopo, North West), and Zimbabwe [3]. Occurrences in the study area are shown in Figure 2b. Chorotype: Southern-Eastern African (SEA).

Ecology: Adults were collected on plants of *Searsia* spp. (Anacardiaceae), *Ziziphus* (Rhamnaceae), and accidentally on *Acacia* shrubs (Fabaceae) at elevations between 900 and 1300 m a.s.l., mainly from October to December [3]. Based on our analysis, *C. luteomaculata* appears as closely associated with the savannah environment (Figure 4), with five occurrence sites all falling within this type of vegetation and characterized by the following mean values for the seven bioclimatic variables considered: BIO2 = 14.92 °C, BIO3 = 54.84%, BIO8 = 23.50 °C, BIO9 = 13.23 °C, BIO13 = 114 mm, BIO14 = 3.60 mm, and BIO15 = 81.22 mm (Figure 5).

*Calotheca luteotessellata* D’Alessandro, Iannella, Grobbelaar, Biondi, 2020 [3]

Distribution: Republic of South Africa (Limpopo) [3] (Figure 2c). Endemic to the study area. Chorotype: Southern-Eastern African (SEA).

Ecology: Adults were collected on *Searsia magaliesmontana* subsp. *coddii* (Anacardiaceae) at elevations between 700 and 900 m a.s.l., in February [3]. Based on our analysis, the only two occurrence sites are both in the savannah environment (Figure 4) and characterized by the following mean values for the seven bioclimatic variables considered: BIO2 = 11.66 °C, BIO3 = 59.63%, BIO8 = 23.60 °C, BIO9 = 16.77 °C, BIO13 = 190 mm, BIO14 = 10 mm, and BIO15 = 86.26 mm (Figure 5).

*Calotheca marginalis* (Weise, 1902) [26]

Distribution: Democratic Republic of the Congo, Eritrea, Ethiopia, Kenya, Republic of South Africa (North West), Rwanda, Somalia, Sudan, South Sudan, Tanzania, and Uganda [2]. In the study area, *C. marginalis* is only recorded from North West Province, Dinokana, 1300 m a.s.l. (Figure 2b). Chorotype: Eastern African (EAF).

Ecology: Adults were collected in Kenya on *Searsia* (sub *Rhus*) *vulgaris*, in June [2,9]. Based on our analysis, the only known locality for *C. marginalis* in the study area falls in a savannah environment (Figure 4), and it is characterized by the following mean values for the seven bioclimatic variables considered: BIO2 = 15.31 °C, BIO3 = 55.26%, BIO8 = 23.30 °C, BIO9 = 12.23 °C, BIO13 = 109 mm, BIO14 = 1 mm, and BIO15 = 81.65 mm (Figure 5).

*Calotheca marmorata* (Baly, 1865) [27]

Distribution: Republic of South Africa (KwaZulu-Natal, Limpopo) [6] (Figure 2b). Endemic to the study area. Chorotype: Southern–Eastern Afrotropical (SEA).

Ecology: Adults were collected on plants of *Searsia* (sub *Rhus*) *pyroides* var. *pyroides* (Anacardiaceae) at elevations between 40 and 1400 m a.s.l. from October to April [6]. Based on our analysis, *C. marmorata* shows clear preference mainly for Grassland and Indian ocean coastal belt vegetation (Figure 4). Its twenty occurrences sites show the following mean values for the seven bioclimatic variables considered: BIO2 = 11.36 °C, BIO3 = 56.83%, BIO8 = 21.75 °C, BIO9 = 14.54 °C, BIO13 = 143.05 mm, BIO14 = 21.45 mm, and BIO15 = 58.51 mm (Figure 5).

*Calotheca natalensis* (Jacoby, 1895) [25]

Distribution: Mozambique, Republic of South Africa (Gauteng, KwaZulu-Natal, Limpopo), Tanzania, and Zambia [2]. Occurrences in the study area are shown in Figure 2b. Subendemic to the study area. Chorotype: Southern-Eastern Afrotropical (SEA).

Ecology: Adults were collected in the study area on *Searsia* (sub *Rhus*) sp., from October to December, mainly at elevations between 600 and 1600 m a.s.l. [2].

Based on our analysis, *C. natalensis* is mainly associated with Savannah and Grassland environments (Figure 4). The nine occurrence sites are characterized by the following mean values for the seven bioclimatic variables considered: BIO2 = 13.79 °C, BIO3 = 56.03%, BIO8 = 21.93 °C, BIO9 = 12.95 °C, BIO13 = 122.78 mm, BIO14 = 8.11 mm, and BIO15 = 72.98 mm (Figure 5).

*Calotheca nigromaculata* (Jacoby, 1888) [28]

Distribution: Mozambique, Namibia, Tanzania, and Republic of South Africa (KwaZulu-Natal, Limpopo, Mpumalanga, Western Cape) [11]. Occurrences in the study area are shown in Figure 2a. Chorotype: probably Southern-Eastern African (SEA).

Ecology: Adults were collected on *Searsia* sp., *S. leptodictya* (along with larvae), *S.* cf. *gueinzii*, *S. pentheri* (Anacardiaceae), and *Allophylus decipiens* (Sapindaceae) at elevations between 40 and 1650 m a.s.l. from October to March [11]. Based on our analysis, *C. nigromaculata* shows clear preferences for Savannah and Grassland environments (Figure 4). In the study area, this species is known with 55 occurrence sites characterized by the following mean values for the seven bioclimatic variables considered: BIO2 = 11.99 °C, BIO3 = 57.67%, BIO8 = 22.89 °C, BIO9 = 15.52 °C, BIO13 = 134.44 mm, BIO14 = 16.47 mm, and BIO15 = 64.31 mm (Figure 5).

*Calotheca nigrotessellata* (Baly, 1865) [27]

Distribution: Botswana, Namibia, Republic of South Africa (Eastern Cape, Free State, Gauteng, KwaZulu-Natal, Limpopo, Mpumalanga, Northern Cape, North West, Western Cape), Zambia, and Zimbabwe [3]. Occurrences in the study area are shown in Figure 2a. Chorotype: Southern African (SAF).

Ecology: Adults were collected on *Searsia leptodictya*, *S. pyroides*, *S. magalismontana* ssp. *coddii*, *S.* cf. *rehmanniana* (Anacardiaceae), and accidentally on *Acacia* sp., *A. nilotica* (Fabaceae), and *Ziziphus* sp. (Rhamnaceae) at elevations between 10 and 1700 m a.s.l. from January to May and from July to December [3]. Based on our analysis, *C. nigrotessellata* results are mainly associated with the savannah environment (Figure 4). This species, with nearly 100 occurrence sites, is the *Calotheca* species and is the most common in the study area. The mean values for the seven bioclimatic variables considered are as follows: BIO2 = 13.79 °C, BIO3 = 55.97%, BIO8 = 20.76 °C, BIO9 = 13.55 °C, BIO13 = 105.08 mm, BIO14 = 11.32 mm, and BIO15 = 65.34 mm (Figure 5).

*Calotheca oberprieleri* D’Alessandro, Iannella, Grobbelaar, Biondi, 2021 [4]

Distribution: Republic of South Africa (Eastern Cape) [4] (Figure 2c). Endemic to the study area. Chorotype: Southern-Western African (SWA).

Ecology: Adults were collected at elevations between 280 and 900 m a.s.l., in November; no information is available on its host plants [4]. Based on our analysis, *C. oberprieleri* is mainly associated with the typical South African vegetation, mainly Albany thicket and Fynbos and, to a lesser extent, with Savannah environment (Figure 4). The four occurrence sites are characterized by the following mean values for the seven bioclimatic variables considered: BIO2 = 10.79 °C, BIO3 = 58.69%, BIO8 = 18.75 °C, BIO9 = 14.64 °C, BIO13 = 81.5 mm, BIO14 = 29 mm, and BIO15 = 33.26 mm (Figure 5).

*Calotheca ornata* (Baly, 1881) [29]

Distribution: Republic of South Africa (Gauteng, KwaZulu-Natal, Limpopo, Mpumalanga, Western Cape) [2] (Figure 2a). Endemic to the study area. Chorotype: Southern African (SAF).

Ecology: adults were collected at elevations mainly between 700 and 2150 m a.s.l. from February to April and from October to December; no information is available on its host plants [2]. Based on our analysis, *C. ornata* shows clear preferences for Grassland and Savannah environments (Figure 4). This species is known with 44 occurrence sites characterized by the following mean values for the seven bioclimatic variables considered: BIO2 = 13.20 °C, BIO3 = 58.25%, BIO8 = 20.63 °C, BIO9 = 12.59 °C, BIO13 = 145.14 mm, BIO14 = 9.68 mm, and BIO15 = 75.32 mm (Figure 5).

*Calotheca orophila* D’Alessandro, Grobbelaar, Iannella, Biondi, 2023 [6]

Distribution: Republic of South Africa (KwaZulu-Natal) [6] (Figure 2c). Endemic to the study area. Chorotype: Southern-Eastern African (SWA).

Ecology: This species was collected in March, but no information is available on its host plants [6]. Based on our analysis, *C. orophila* appears associated with high-altitude Grassland (Figure 4). This species is known from a single locality in KwaZulu-Natal (Sani Pass, 1890 m a.s.l.), characterized by the following mean values for the seven bioclimatic variables considered: BIO2 = 11.28 °C, BIO3 = 52.48%, BIO8 = 12.23 °C, BIO9 = 3.65 °C, BIO13 = 140 mm, BIO14 = 12 mm, and BIO15 = 68.51 mm (Figure 5).

*Calotheca pallida* (Bryant, 1945) [30]

Distribution: Republic of South Africa (Western Cape) [4] (Figure 2b). Endemic to the study area. Chorotype: Southern-Western African (SWA).

Ecology: Adults were collected at elevations between 30 and 70 m a.s.l., in January, April and December [4]; no information is available on its host plants. Based on our analysis, *C. pallida* is closely associated with Fynbos (Figure 4). Its 2 known occurrence sites are characterized by the following mean values for the seven bioclimatic variables considered: BIO2 = 10.69 °C, BIO3 = 58.07%, BIO8 = 15.37 °C, BIO9 = 21.11 °C, BIO13 = 48 mm, BIO14 = 28.50 mm, and BIO15 = 16.09 mm (Figure 5).

*Calotheca parvula* (Weise, 1908) [31]

Distribution: Namibia (Karas), and Republic of South Africa (Eastern Cape, Northern Cape, Western Cape) [4]. Occurrences in the study area are shown in Figure 2b. Chorotype: Southern–Western African (SWA).

Ecology: Adults were collected in the study area on *Searsia* spp. (Anacardiaceae) at elevations between 700 and 750 m a.s.l. from September to July [4]. Based on our analysis, *C. parvula* is mainly associated with Nama karoo, Succulent karoo, and Fynbos (Figure 4). In the study area, this species is known from nine occurrence sites, characterized by the following mean values for the seven bioclimatic variables considered: BIO2 = 15.56 °C, BIO3 = 54%, BIO8 = 17.85 °C, BIO9 = 16.48 °C, BIO13 = 47.67 mm, BIO14 = 7.22 mm, and BIO15 = 55.13 mm (Figure 5).

*Calotheca pauli* (Weise, 1905) [32]

Distribution: Kenya, Mozambique, Republic of South Africa, and Zimbabwe. Known only from KwaZulu-Natal (Ndumo Game Reserve), this species is recorded for the first time in the study area (Figure 2b). Chorotype: Eastern Afrotropical (EAF)

Ecology: No information is available on the host plants of this species. Adults were collected in the study area, at an elevation of 80 m a.s.l., in January [2]. From our analysis, the only known locality for *C. pauli* in the study area falls in Savannah environment (Figure 4), characterized by the following mean values for the seven bioclimatic variables considered: BIO2 = 12.49 °C, BIO3 = 59.20%, BIO8 = 26.38 °C, BIO9 = 19.10 °C, BIO13 = 106 mm, BIO14 = 9 mm, and BIO15 = 66.48 mm (Figure 5).

*Calotheca prinslooi* D’Alessandro, Iannella, Grobbelaar, Biondi, 2021 [4]

Distribution: Republic of South Africa (Eastern Cape, Western Cape) [4] (Figure 2c). Endemic to the study area. Chorotype: Southern-Western African (SWA).

Ecology: adults were collected on plants of *Searsia pallens* and *S. dentata* (Anacardiaceae), at elevations between 100 and 900 m a.s.l., from January to May and from October to December [4]. Based on our analysis, *C. prinslooi* is mainly associated with Fynbos and Albany thicket (Figure 4). The 18 occurrence sites are characterized by the following mean values of the seven bioclimatic variables considered: BIO2 = 11.68 °C, BIO3 = 55.03%, BIO8 = 17.45 °C, BIO9 = 16.67 °C, BIO13 = 56 mm, BIO14 = 27.06 mm, and BIO15 = 22.83 mm (Figure 5).

*Calotheca regularis* (Jacoby, 1900) [33]

Distribution: Republic of South Africa (Western Cape) [2] (Figure 2b). Endemic to the study area. Chorotype: Southern–Western African (SWA).

Ecology: Adults were collected at elevations between 10 and 350 m a.s.l., in December and January; no information is available on its host plants [2]. Based on our analysis, *C. regularis* mainly shows preferences for forest environment and, to a lesser extent, for Fynbos and Albany thicket (Figure 4). The seven occurrence sites are characterized by the following mean values for the seven bioclimatic variables considered: BIO2 = 10.36 °C, BIO3 = 55.62%, BIO8 = 18.55 °C, BIO9 = 14.42 °C, BIO13 = 78 mm, BIO14 = 38.29 mm, and BIO15 = 19.38 mm (Figure 5).

*Calotheca reticulata* (Baly, 1865) [27]

Distribution: Malawi, Republic of South Africa (Gauteng, Limpopo, North West, Western Cape), Tanzania, and Zimbabwe [2]. Occurrences in the study area are shown in Figure 2b. Chorotype: possibly Eastern Afrotropical (EAF).

Ecology: Adults of this species were mainly collected at elevations between 1100 and 1400 m a.s.l. from November to February; no information is available on its host plants [2]. Based on our analysis, *C. reticulata* appears mainly associated with Savannah environment in the study area (Figure 4). The seven occurrence sites in the study area are characterized by the following mean values for the seven bioclimatic variables considered: BIO2 = 13.59 °C, BIO3 = 56.11%, BIO8 = 21.84 °C, BIO9 = 12.76 °C, BIO13 = 137.43 mm, BIO14 = 5.43 mm, and BIO15 = 81.61 mm (Figure 5).

*Calotheca thunbergi* Biondi and D’Alessandro, 2017 [2]

Distribution: Republic of South Africa (Eastern Cape, Western Cape) [5] (Figure 2c). Endemic to the study area. Chorotype: Southern-Western African (SWA).

Ecology: Adults were collected on *Searsia* (sub *Rhus*) *dentata* (Anacardiaceae), mainly at elevations between 100 and 1200 m a.s.l. from September to December [5]. Based on our analysis, *C. thunbergi* is mainly associated with the typical South African vegetation, such as Albany thicket and Fynbos, more rarely with forest environments (Figure 4). The seven occurrence sites are characterized by the following mean values for the seven bioclimatic variables considered: BIO2 = 11.82 °C, BIO3 = 56.80%, BIO8 = 19.64 °C, BIO9 = 13.62 °C, BIO13 = 59.14 mm, BIO14 = 26.43 mm, and BIO15 = 26.02 mm (Figure 5).

*Calotheca vittata* (Baly, 1862) [34]

Distribution: Mozambique, and Republic of South Africa (KwaZulu-Natal) [2]. Occurrences in the study area are reported in Figure 2b. Subndemic to the study area. Chorotype: Southern–Eastern African (SEA).

Ecology: Adults were mainly collected in the study area at elevations between 10 and 1050 m a.s.l. from April to June and from October to November; no information is available on its host plants [2]. From our analysis, *C. vittata* results mainly associated with Indian ocean coastal belt vegetation (Figure 4). The 14 occurrence sites are characterized by the following mean values for the seven bioclimatic variables considered: BIO2 = 10.47 °C, BIO3 = 56.53%, BIO8 = 23.48 °C, BIO9 = 17.18 °C, BIO13 = 140.14 mm, BIO14 = 36.21 mm, and BIO15 = 42.79 mm (Figure 5).

*Calotheca wanati* D’Alessandro, Iannella, Grobbelaar, Biondi, 2022 [11]

Distribution: Republic of South Africa (KwaZulu-Natal) [11] (Figure 2c). Endemic to the study area. Chorotype: Southern-Eastern African (SEA).

Ecology: Adults were collected on plants of *Allophylus natalensis* (Sapindaceae) in November, December, and January at elevations between 5 and 500 m a.s.l. [11]. Based on our analysis, *C. wanati* shows clear preferences for Savannah environment (Figure 4). The 4 occurrence sites are characterized by the following mean values for the seven bioclimatic variables considered: BIO2 = 10.90 °C, BIO3 = 57.50%, BIO8 = 24.24 °C, BIO9 = 17.31 °C, BIO13 = 114.25 mm, BIO14 = 15.50 mm, and BIO15 = 56.80 mm (Figure 5).

### 3.2. Distribution of the Calotheca Species in the Study Area

Assuming a more or less homogeneous sampling effort throughout the study area, seventeen of the twenty-five species (68%) occur in fewer than eleven sites (Figure 6 and Figure 7), and nine of them have been reported in fewer than six sites. In contrast, four species, *Calotheca haroldi*, *C. nigromaculata*, *C. nigrotessellata*, and *C. ornata*, show a significantly higher frequency, from forty to about one hundred occurrence sites.

Fourteen species (56%) are endemic to the study area (Figure 6 and Figure 7).

Of these, *Calotheca leonardii*, *C. marmorata*, and *C. prinslooi* occur in 20, 17 and 18 localities, while the remaining, *C. carolineae*, *C. danielssoni*, *C. inornata*, *C. luteotessellata*, *C. oberprieleri*, *C. ornata*, *C. orophila*, *C. parvula*, *C. regularis*, *C. thunbergi*, and *C. wanati*, are among the species with less than 10 occurrence sites (Figure 7).

The endemic species are all distributed in the areas of highest suitability for the genus (Figure 8), mainly the north-eastern region, the long coastal and sub-coastal area of the Indian Ocean (Richards Bay to Mossel Bay), and the south-western Atlantic coastal region. Areas of lower suitability are instead generally occupied by species with a wider distribution and associated with more vegetation types, such as *C. nigrotessellata*, or by species with a preference for peculiar vegetation types, such as *C. parvula*. *Calotheca nigrotessellata*, which is clearly eurytopic, is present in 9 different vegetation types (Figure 4) and shows a greater preference for the savannah environment; *C. parvula*, although representing a species associated with the typical South African vegetation, is the only one that extends significantly in the most internal areas of Nama Karoo, an environment considered by our analysis as non-optimal for the presence of *Calotheca*.

### 3.3. Calotheca Species, Bioclimatic Variables, Vegetation Types, and Host Plants

From a climatic point of view (Figure 5), the most significant variables to describe the distribution of the species of the genus *Calotheca* in the study area are as follows: BIO8, mean temperature of wettest quarter; BIO9, mean temperature of driest quarter; BIO13, precipitation of wettest period. As regards the variable BIO8, the lowest mean values are characteristic of the occurrence sites of species associated with high-altitude grasslands (BIO8 < 13 °C), such as *C. orophila*; medium mean values (13 < BIO8 < 20 °C) are generally found in the occurrence sites of species with south-western distribution associated with Fynbos, Succulent and Nama karoo, Albany thicket; higher mean values (BIO8 > 20 °C) are found in the occurrence sites of species with north-eastern distribution associated with Grassland, Savannah, and Forest, as, for example, *C. carolineae*, *C. pauli*, and *C. wanati*. The temperature BIO9 shows an opposite behaviour compared to BIO8, apart from the similarly low value in high-altitude grasslands; on average, it has the lowest values in the occurrence sites falling in the Savannah (BIO9 < 16 °C) and the highest for those sites associated with Fynbos (16 < BIO9 < 22 °C). The precipitation variable BIO13 shows a clear dissimilarity between the occurrence sites in environments with typical South African vegetation (BIO13 < 100 mm), a characteristic condition of the species of the south-western sector mainly associated with Albany thicket and Fynbos, such as *C. danielssoni*, *C. pallida*, *C. parvula*, *C. prinslooi*, and *C. thunbergi*, and the occurrence sites in Grassland and Savannah environments (BIO13 > 100 mm), occupied instead by the species of the north-eastern sector, as, for example, *C. luteotessellata*, *C. marmorata*, *C. reticulata*, and *C. vittata*.

With respect to vegetation (Figure 4), the genus *Calotheca* in the study area can be divided into three large groups: (a) species exclusively or preferentially associated with Savannah and Grassland, mainly with eastern distribution, such as *C. haroldi*, *C. holubi*, *C. luteomaculata*, *C. luteotessellata*, *C. marginalis*, *C. nigromaculata*, *C. nigrotessellata*, *C. ornata*, *C. orophila*, *C. pauli*, *C. reticulata*, and *C. wanati*; (b) species exclusively or preferentially associated with forest environments, mainly with south-eastern distribution, such as *C. carolineae*, *C. marmorata*, *C. natalensis,* and *C. vittata*; (c) species associated with the typical South African vegetation, mainly Albany thicket, Fynbos, Succulent karoo, and Nama karoo, with western distribution, such as *C. danielssoni*, *C. inornata*, *C. leonardii*, *C. oberprieleri*, *C. pallida*, *C. parvula*, *C. prinslooi*, *C. regularis*, and *C. thunbergi*.

Regarding fourteen endemic species, the association with the vegetation types is quite varied (Figure 4). In fact, we have four species significantly associated with Savannah and/or Grassland: *C. luteotessellata*, *C. marmorata*, *C. ornata*, and *C. wanati*; but also, eight species associated with the typical South African vegetation: *C. danielssoni*, *C. inornata*, *C. pallida*, *C. prinslooi*, and *C. regularis*, mainly with preference for Fynbos, and *C. leonardii*, *C. oberprieleri*, and *C. thunbergi*, associated with Succulent karoo vegetation. The only species found to be associated exclusively with the coastal forest environment is the endemic *C. carolineae*.

The cluster analysis (Figure 9), apart from the couple *holubi-marginalis*, exclusively associated with Savannah and characterized by low values of BIO14 (precipitation of driest period), and the isolated position of *orophila*, with a very low value of BIO9 (mean temperature of driest quarter), substantially identified the following three clusters: (1) *carolinae-vittata-inornata-nigromaculata-wanati*, with species generally extending to forest environments, with medium–low values of BIO2 (mean diurnal range), medium values of BIO14, and high values of BIO13 (precipitation of wettest period); (2) *haroldi-reticulata-luteomaculata-luteotessellata-natalensis-ornata-nigrotessellata-pauli*, with species mainly associated with Savannah, with medium values of BIO2, low values of BIO14, and high values of both BIO15 (precipitation seasonality) and BIO13; (3) *leonardii-oberprieleri-prinslooi-thunbergi-regularis-pallida*, with endemic species with western distribution in the study area, associated with Albany ticket, with comparatively lower values of BIO 13 and BIO15. Finally, the analysis identified the couple *danielssoni*-*parvula*, both occurring in Succulent karoo, with medium–low values of BIO14 and medium–high of BIO13.

About *Searsia* plants, the map in Figure 10 shows the results obtained by the density-based clustering analysis, which identified the areas of higher-density occurrence within the study area. The six following clusters were found: cluster 1, which contains 1247 sites of the 18,208 total (6.85%), is located in a wide area in southern KwaZulu-Natal, and includes the coastal region from Richards Bay to Port Shepstone, and the internal areas up to the Drakensberg; cluster 2 (1400 sites, 7.69%) shows the location furthest from the coast and presents the province of Gauteng as the central area; cluster 3 (534 sites, 2.93%) is the northernmost, mainly including the Mpumalanga plateaus; clusters 4–6, with 1528 sites (8.39%), 1664 (9.14%), and 6308 (34.65%), respectively, are the most numerically representative in the study area and are located, from east to west, in the southernmost coastal and sub-coastal regions, from the south-western Eastern Cape to the southern Western Cape up to the Cape Peninsula. From a vegetation point of view, cluster 1 is mainly associated with Indian Ocean Coastal Belt, Grassland and Highveld Grassland, while clusters 2–3 are mainly associated with Savannah. Finally, clusters 4–6 are closely associated with typical South African vegetation, particularly Fynbos and, to a lesser extent, Albany thicket. Comparing the map in Figure 10 with the distribution of the suitability areas for the genus *Calotheca* (Figure 8), the six different clusters identified for *Searsia* are largely overlapping with the areas of higher suitability that our model returned for *Calotheca*.

## 4. Discussion

South Africa, with its 25 species, hosts around 68% of the total *Calotheca* known so far (37), of which 14, equal to 56% of the species found in South Africa and 38% of the total species, are endemic.

Considering a boundary line represented from west to east by the course of the Orange River, which then continues with that of the Tugela River, in correspondence with the Eastern Escarpment, the study area can be virtually divided into a north-eastern area and one south-western of similar extension. The north-eastern area, in particular in the North West, Limpopo, and Mpumalanga provinces, is characterized by the presence of *Calotheca* species with a wider distribution in sub-Saharan Africa and mainly associated with Savannah, such as *C. haroldi*, *C. holubi*, *C. marginalis*, and *C. pauli*, while others are endemic, although showing close affinities with the sub-Saharan species with an eastern distribution. Among these, *C. luteotessellata* from Limpopo, and *C. wanati* and *C. carolineae*, from north-eastern KwaZulu-Natal. The south-western sector hosts instead the species associated with the typical South African vegetation, most of which are endemic, such as *C. leonardii*, *C. oberprieleri*, *C. thunbergi* mainly associated with the Albany thicket, and *C. danielssoni*, *C. prinslooi*, *C. inornata* associated with the Fynbos. Only *Calotheca regularis* and the common *C. nigrotessellata* are occurring in both sectors. Other peculiar distributions concern: *Calotheca orophila*, the only species linked to the high-altitude Grasslands, and *Calotheca parvula*, the only species that extends to the Nama karoo, an environment that does not host any other species of *Calotheca*.

To summarize our findings about the bioclimatic variables, temperature and precipitation emerge as critical drivers, influencing the habitat suitability of different species. For instance, *C. orophila*, a species restricted to high-altitude grasslands, exemplifies the adaptation to cooler and wetter conditions compared to other species, such as *C. wanati* and *C. carolineae*, which thrive in warmer climates and more humid environments. The contrasting ecological niches found underline the genus’s capacity to adapt to a wide range of thermal and moisture conditions. From a vegetation perspective, *Calotheca* species can be grouped based on their associations with distinct biomes, such as for *C. haroldi* and *C. nigrotessellata*, primarily linked to savannah and grassland ecosystems of the eastern parts of the study area. Conversely, species like *C. danielssoni* and *C. thunbergi* are more closely tied to the unique vegetation types of South Africa, including Fynbos and Albany thicket, which are characteristic of the western region. This geographical partitioning reflects the interplay between vegetation type and climatic gradients, underlining the role of both biotic and abiotic factors in shaping species distribution.

Because of these two ecological aspects found, endemism within the genus is confirmed to be particularly notable, with over half of the species in the study area exhibiting restricted distributions. These endemic species demonstrate a marked preference for specialized habitats, often aligned with South Africa’s unique vegetation, such as Succulent Karoo and coastal forests. This highlights the conservation importance of these habitats, which harbour a significant portion of the region’s biodiversity.

Also, the cluster analysis further supports these observations, grouping species based on shared ecological and climatic traits. Notably, species associated with savannah environments form distinct clusters characterized by greater temperature variability and seasonal precipitation patterns. In contrast, species linked to Fynbos or Albany thicket display adaptations to more stable climatic conditions, with relatively moderate temperature and precipitation ranges. These groupings provide relevant information about the ecological strategies adopted by *Calotheca* species to exploit their respective niches.

Finally, our results highlighted that the distribution of *Calotheca* in the study area is strongly related with that of the genus *Searsia*, which includes most of the host plants for this flea beetle genus. In fact, the areas that our model has identified as the most suitable for the presence of the *Calotheca* species are also those where the density of *Searsia* is highest.

## 5. Conclusions

The goal of this paper, providing an ecological profile, albeit preliminary, for the individual *Calotheca* species and the entire genus, was possible by the accuracy of recently acquired taxonomic, distributional, autoecological, and climatic data. These updates have enabled the implementation of an increasingly informative database, which, despite its chronic incompleteness, has been the basic tool for the analyses reported in this study.

In fact, through a methodological approach which used proper analyses, it was possible to identify a hitherto unavailable possible ecological and biogeographical framework and the individual factors that influence species distribution. These data are required for each kind of investigation on the target species, particularly for taxa and areas that lack historical taxonomic and faunistic background.

## Figures and Tables

**Figure 1 insects-15-00994-f001:**
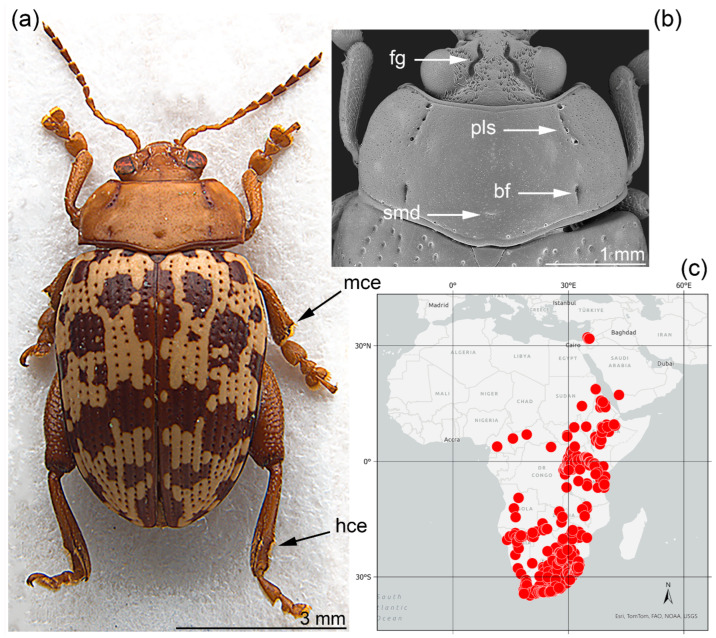
Diagnostic characters and distribution of *Calotheca* Heiden: (**a**) habitus of *C. carolineae* D’Alessandro, Iannella, Grobbelaar, Biondi, 2022 [11], modified from D’Alessandro et al., 2022 [11]; (**b**) head and pronotum of *C. nigromaculata* (Jacoby), modified from D’Alessandro et al., 2022 [11]; (**c**) distribution of *Calotheca* (red dots). bf: basal furrow; fg: frontal groove; pls: punctate lateral stria; smd: shallow medial dimple.

**Figure 2 insects-15-00994-f002:**
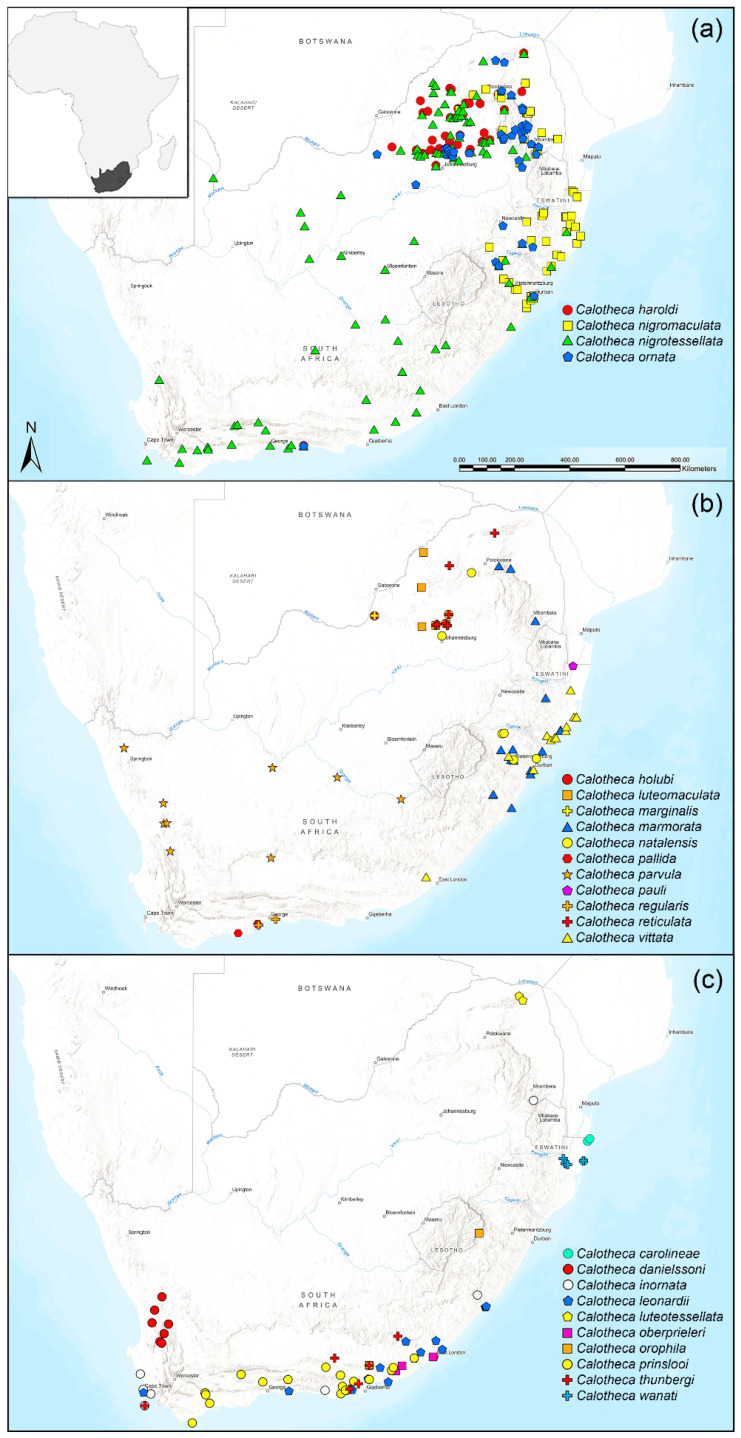
Distribution of Calotheca species in the study area: (**a**) Calotheca haroldi, *C. nigromaculata*, *C. nigrotessellata*, and *C. ornata*; (**b**) ditto of *C. holubi*, *C. luteomaculata*, *C. marginalis*, *C. marmorata*, *C. natalensis*, *C. pallida*, *C. parvula*, *C. pauli*, *C. regularis*, *C. reticulata*, and *C. vittata*; (**c**) ditto of *C. carolineae*, *C. danielssoni*, *C. inornata*, *C. leonardii*, *C. luteotessellata*, *C. oberprieleri*, *C. orophila*, *C. prinslooi*, *C. thunbergi*, and *C. wanati*.

**Figure 3 insects-15-00994-f003:**
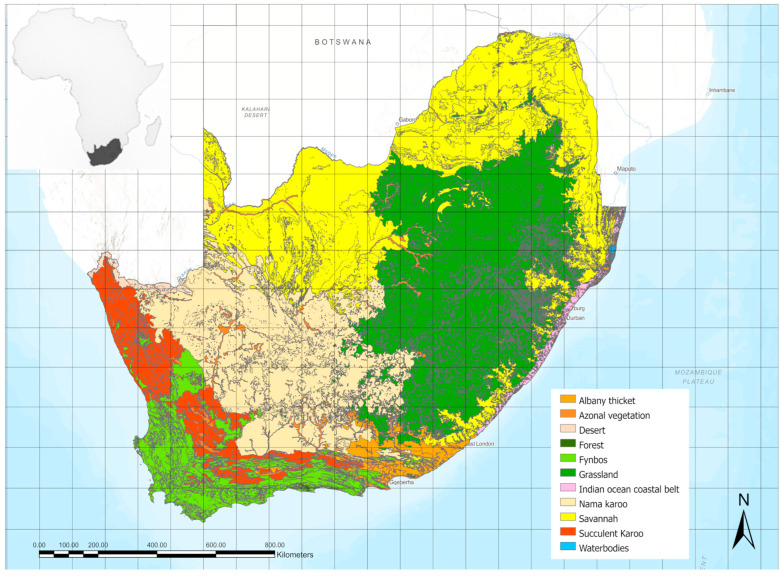
Map of the vegetation types (biomes) occurring in the study area.

**Figure 4 insects-15-00994-f004:**
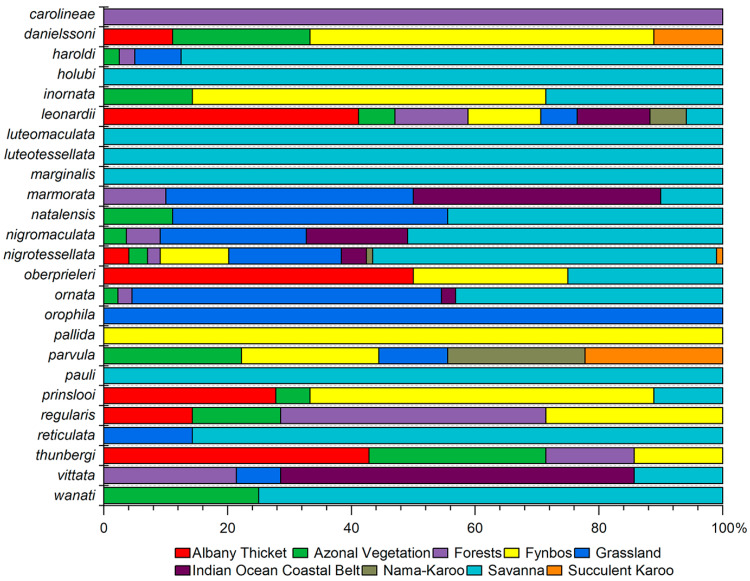
Presence percentages of the 25 *Calotheca* species in the vegetation types.

**Figure 5 insects-15-00994-f005:**
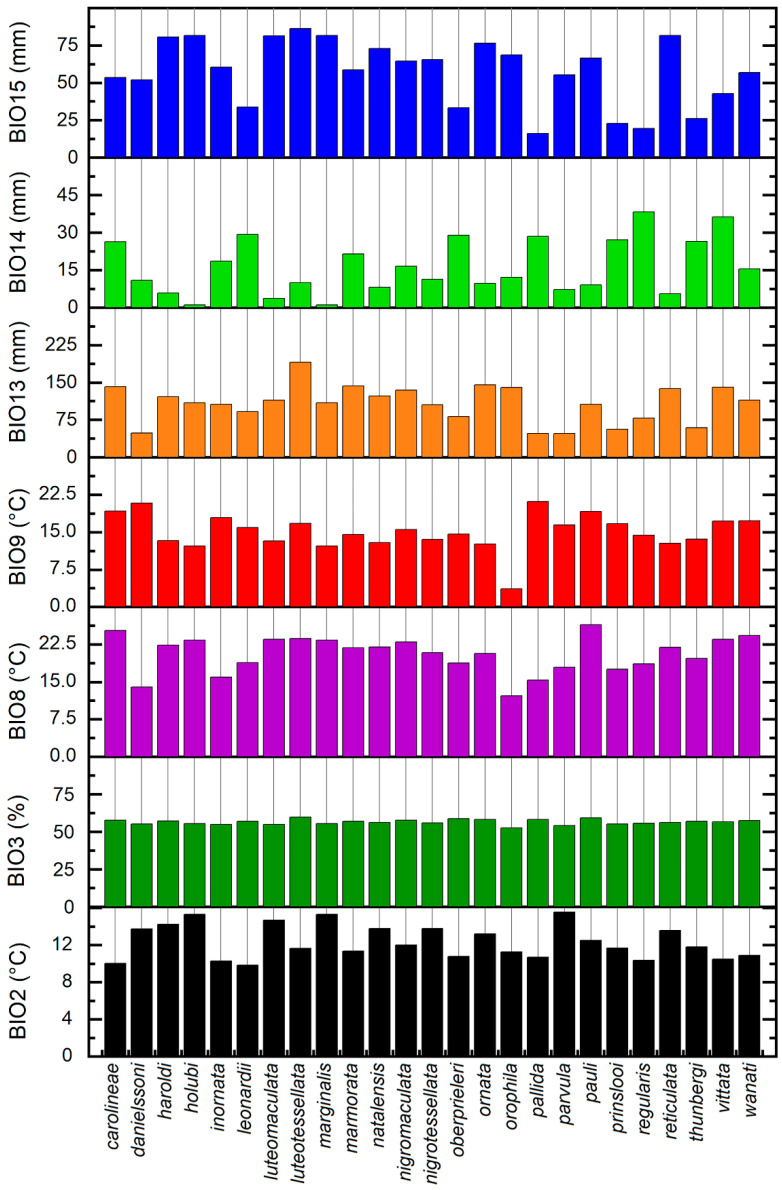
Mean values of bioclimatic variables for the 25 *Calotheca* species.

**Figure 6 insects-15-00994-f006:**
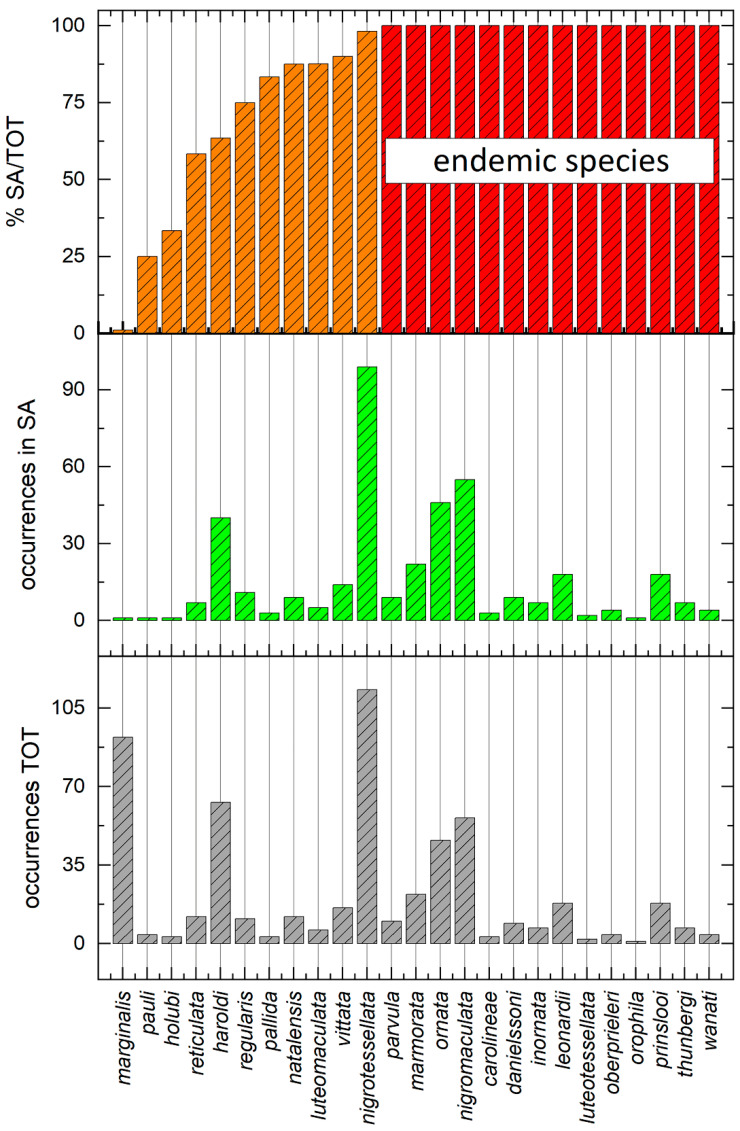
For each of the 25 *Calotheca* species: total number of occurrence sites known; total number of occurrence sites in the study area; percentage of occurrence sites in the study area compared to the total number of occurrence sites.

**Figure 7 insects-15-00994-f007:**
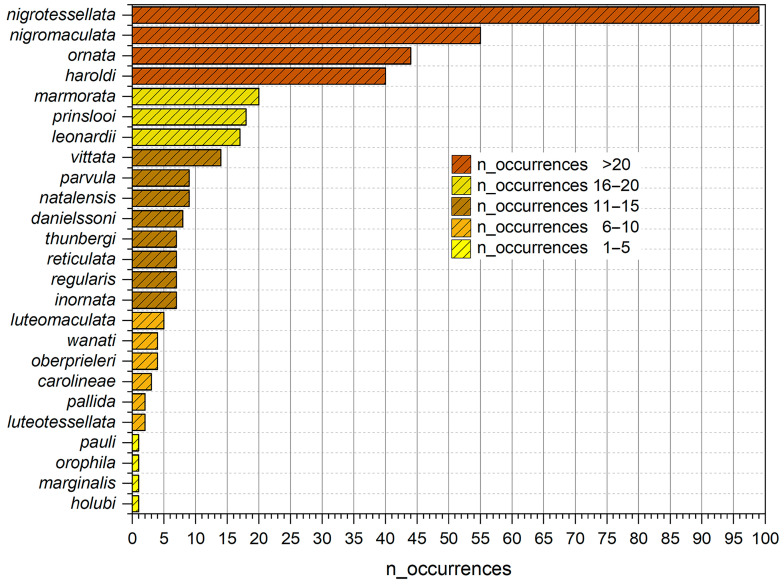
Number of occurrence sites for each of the 25 *Calotheca* species in the study area.

**Figure 8 insects-15-00994-f008:**
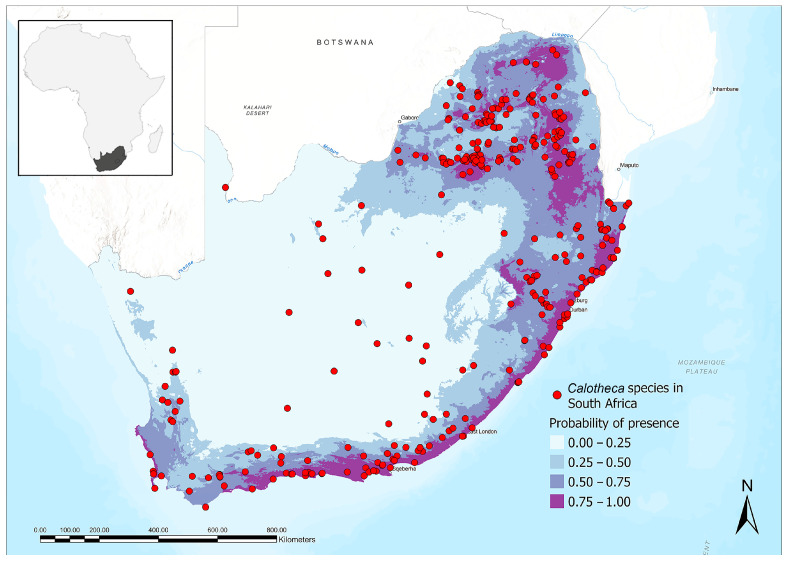
Distribution and areas of suitability of the genus *Calotheca* in the study area.

**Figure 9 insects-15-00994-f009:**
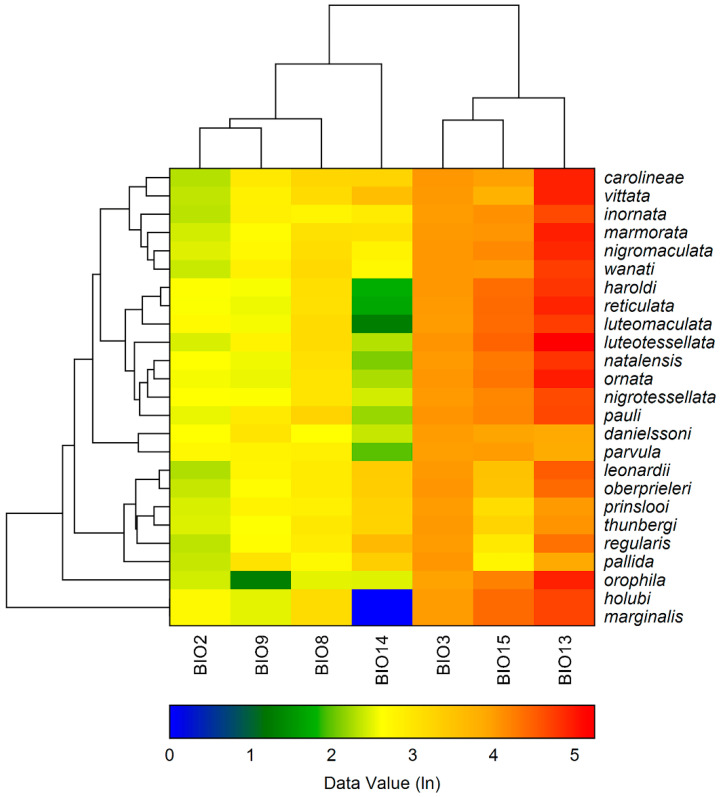
Double clustered heat maps species/bioclimatic variables (see text).

**Figure 10 insects-15-00994-f010:**
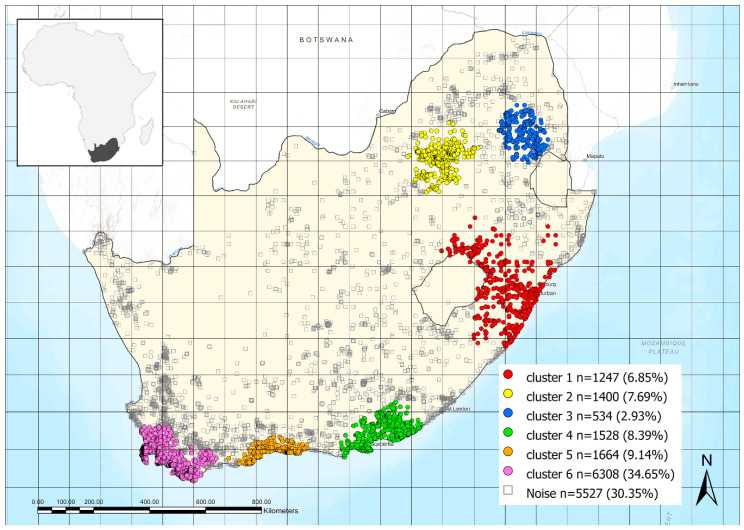
Density-based clustering with HDBSCAN method for the occurrence sites of the plant genus *Searsia* (Anacardiaceae) (see text).

**Table 1 insects-15-00994-t001:** Coding and explanation of the bioclimatic variables used, from the Worldclim.org repository.

Variable Code	Variable Description
BIO01	Annual Mean Temperature
BIO02	Mean Diurnal Range (Mean of monthly (max temp − min temp))
BIO03	Isothermality (BIO02/BIO07) (×100)
BIO04	Temperature Seasonality (standard deviation ×100)
BIO05	Max Temperature of Warmest Month
BIO06	Min Temperature of Coldest Month
BIO07	Temperature Annual Range (BIO05-BIO06)
BIO08	Mean Temperature of Wettest Quarter
BIO09	Mean Temperature of Driest Quarter
BIO10	Mean Temperature of Warmest Quarter
BIO11	Mean Temperature of Coldest Quarter
BIO12	Annual Precipitation
BIO13	Precipitation of Wettest Month
BIO14	Precipitation of Driest Month
BIO15	Precipitation Seasonality (Coefficient of Variation)
BIO16	Precipitation of Wettest Quarter
BIO17	Precipitation of Driest Quarter
BIO18	Precipitation of Warmest Quarter
BIO19	Precipitation of Coldest Quarter

## Data Availability

Upon request, the authors can provide the original data used in this paper.

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
