# Peer review of "Ecological Profile of the Flea Beetle Genus Calotheca Heyden in South Africa (Chrysomelidae, Galerucinae, Alticini)"

_insects, 2024, doi:10.3390/insects15120994_

Round 1
Reviewer 1 Report
Comments and Suggestions for Authors
I have made a number of New Comments directly onto the Review copy.

Author Response
Reviewer 1
Dear Reviewer, thank you for your comments. We addressed all of them, as reported below. The line numbering refers to the track-changes version of the manuscript.
Commented [A1] Why is this analysis only for the Calotheca from South Africa? Figure 1C shows the complete distribution throughout all of Africa. Wouldn’t an analysis of the entire genus be more informative? It would only be adding the analysis for another 12 species!
We have added the following paragraph to the text (lines 65-71):
“Our choice to consider only South Africa as reference area is based on the following considerations: a) it is a well-defined area from a biogeographical point of view, where the biodiversity of the genus Calotheca is at its maximum; b) this area is characterized by a well-distributed sampling effort, effort that is decidedly less in other areas of the continental Africa affected by the presence of this flea beetle genus; c) availability of environmental data for this area, in particular vegetation data, which made possible many of the analyses we carried out.”
Commented [A2] It is interesting that for 11 of the 25 (i.e., 44%) species treated here there is no host plant information. Doesn’t that effect the analyses in some ways? Or should there be comments about this?
The missing data about the host plants for 11 of 25 species did not influence the analyses performed, as none of the analyses were based on the host plants of the single Calotheca species.
Commented [A3] Were these from the regular USNM NMNH collection or specimens provided from D. Furth there?
Corrected. They are both specimens provided from D. Furth and specimens from the regular USNM NMNH collection. “DGF: Collection of D.G. Furth, …..” was added to the list
Commented [A4] Why is this reference listed this way instead like the rest as a reference number?
Corrected.
Commented [A5] What about Azonal vegetation which is indicated on figure 4 as being a significance component of this species’ distribution? I realize that “Azonal vegetation” maybe a combination of other biomes and ecological conditions, but in this case and others below shouldn’t it be mentioned since it may inform regarding the ecology of each species?? There are 13 of the 25 Calotheca species with Azonal vegetation as part of their Ecology!! Especially for species like this species and a few others where Azonal is a big part of their ecological distribution
Regarding this comment, Reviewer 1 was probably misled by the apparent extent of the Azonal vegetation in the map of Fig. 3, where the coloration of the Azonal vegetation was the same as that of the Nama Karoo. We have solved this problem changing the colors of these two vegetation types. We thus changed the Figure 3, replacing it with a new version.
However, since the "azonal vegetation" includes vegetation types not directly influenced by zonal biomes, it was not considered in the analyses, as it is a category of "grey vegetation" strongly influenced by human activities. Therefore, the presence of Calotheca in this type of vegetation should be considered secondary, and therefore does not contribute to defining the ecological profile of the species.
Commented [A6] There are actually 8 vegetation types associated with this species and Albany Thicket is the largest an isn’t even mentioned here. Why?
Corrected. This was an error. Thanks for the report.
Commented [A7] Isn’t this data from Furth & Young, 1988? Even though it was reported in Biondi et 2017 it may be best to cite the original source. Not critical, but..
Corrected. Anyway, the reference was already reported in the general part of the genus Calotheca.
Commented [A8] What about the Albany Thicket that is the largest vegetation type for this species?
Corrected.
Commented [A9] What about the Azonal vegetation type?
Please see the reply in Commented [A5].
Commented [A10] What about Azonal Vegetation?
Please see the reply in Commented [A5].
Commented [A11] What about the Azonal Thicket component of the vegetation for this species?
Please see the reply in Commented [A5].
Commented [A12] What about Azonal?
Please see the reply in Commented [A5].
Commented [A13] Although this is still part of the Results section, it is clearly another subsection and for the reader’s benefit it may be helpful to have a sub-section title here
We have divided into different sections the Results.
Commented [A14] There may be an argument that these 4 paragraphs could be included in the Discussion section?
The four paragraphs are an integral part of the results. So, we prefer to keep them in their original position. However, we added in the Discussion several considerations based on them.
Commented [A15] What is “..typical South African vegetation”?? It would be helpful to understand what is meant by this phrase that is used in several place in this manuscript.
Corrected. We have added lines 158-160 in Materials and Methods.
Commented [A16] But what can you say about the optimum environmental conditions/requirements for Calotheca, e.g., temperature range, precipitation range,
Comments about the optimum environmental/requirements for Calotheca based on vegetation types were already reported in the lines 580-628. Indeed, we added a new part of the Discussion to consider these topics (L 692-719).
Reviewer 2 Report
Comments and Suggestions for Authors
This paper described 25 species of flea beetle genus Calotheca in South Africa. The authors tried to analyze the ecological and biogeographical relationships between those species and their host plants. I don’t think this manuscript can be considered to be published before a major revising because there are quite a lot questions in this version. The biggest question is that this manuscript needs to be re-structured. This manuscript is too long and has to be squeezed. The followings are my suggestions:
1. Line 57 Is it necessary to cite all same authors references here? May No.6 will be enough because 3,4,5 possibly included in this one.
2. Line 80 Similar references here and try to say 25 species??
3. Line 81-98 Suggest to make a table and easy to follow, currently it is quite difficult to understand.
4. Line 132-149 Similar question: those vegetation types can be list in a table and make a summary under the table will be easy to read.
5. Line 157-171 Collect the information into a table, please, because it is difficult to compare currently.
6. Line 181 Reference 19 is not in English and published in 1908, I don’t know why you cited this one.
7. Line 134-148 Should build up a similar table to previous information table because these two are related.
8. Line 155-217 Make a similar table to previous information table, please
9. All those details of 25 species should be moved to supplement part because it is too long in the main part. Suggest to keep critical information in a table to compare.
10. Line 253,343,355,356,495 What are the meaning of question marks in those lines?
11. Line 567-576 Suggest to make a table, please
12. The discussion part is quite similar to the result part, it doesn’t look like to discuss and has to be reorganized. Especially, the last paragraph is quite similar to previous result needs to be rewritten or deleted.
13. All the format of references are not correct because there is no information of issues. Too many really old references, suggest to find some new references.
Comments on the Quality of English LanguageThe writting has to be improved because it is quite 'talktive'. Too much irrelavant details inclued in this version.
Author Response
Reviewer 2
This paper described 25 species of flea beetle genus Calotheca in South Africa. The authors tried to analyze the ecological and biogeographical relationships between those species and their host plants. I don’t think this manuscript can be considered to be published before a major revising because there are quite a lot questions in this version. The biggest question is that this manuscript needs to be re-structured. This manuscript is too long and has to be squeezed. The followings are my suggestions:
R: Dear Reviewer, thank you for your comments. We addressed all of them, as reported below. The line numbering refers to the track-changes version of the manuscript.
- Line 57 Is it necessary to cite all same authors references here? May No.6 will be enough because 3,4,5 possibly included in this one.
Ref. 6 is actually about the description of two new species, and is more recent than the others; thus, it has no chance to be included in the others.
- Line 80 Similar references here and try to say 25 species??
Thank you for your comment. In the Introduction, we address the broader context of the taxonomic issue, emphasizing the historical progress in knowledge, which is crucial to understanding the gap that our work aims to bridge. The mention of the 25 species there sets the stage for the later methodological description.
In line 80, the reference to "25 species" specifically pertains to the dataset used in this study, outlining the methodological framework and sources of the data. While the same number is mentioned, its role in the narrative is distinct.
We have rephrased the sentences to clarify this distinction and ensure there is no redundancy or confusion.
- Line 81-98 Suggest to make a table and easy to follow, currently it is quite difficult to understand.
Thank you for your suggestion. While we understand the intent behind your comment to improve clarity, in taxonomic practice it is not customary to present the list of repositories or museums in table format. This approach is rarely, if ever, used in scientific literature, as these lists are traditionally presented in textual form to maintain a standardized style and facilitate citation practices.
Moreover, our current format aligns with the conventions followed in taxonomic studies, where repository information is often embedded within the text for seamless integration into the narrative.
- Line 132-149 Similar question: those vegetation types can be list in a table and make a summary under the table will be easy to read.
Thank you for your comment. While we appreciate the suggestion to summarize the vegetation types in a table, the detailed prose accompanying each type provides critical contextual and explanatory information that would be lost if condensed into a tabular format. These descriptions are essential to convey the nuances and ecological significance of each vegetation type, which are integral to the discussion and understanding of our work.
We believe that maintaining the current format ensures clarity and preserves the depth of information necessary for readers to fully grasp the complexity of the vegetation types under consideration.
- Line 157-171 Collect the information into a table, please, because it is difficult to compare currently.
Corrected, we moved all the information to Table 1.
- Line 181 Reference 19 is not in English and published in 1908, I don’t know why you cited this one.
Thank you for your observation. Indeed, Reference 19 is not in English because, at the time of its publication in 1908, it was not customary to publish scientific works in English. However, this reference is foundational for the type of spatial analysis conducted in this study. It is a seminal work that laid the groundwork for methodologies still in use today, including those incorporated into the cutting-edge ArcGIS Pro software by ESRI.
This methodology is directly implemented in ESRI's spatial analysis toolbox, and the toolbox documentation itself cites this work as a fundamental reference. For these reasons, it is crucial to include this citation, as it represents both a historical and practical cornerstone of the analysis.
- Line 134-148 Should build up a similar table to previous information table because these two are related.
Thank you for your suggestion. As this comment is closely related to a previous one (Line 132-149), I would refer you to the response provided earlier. We have explained why summarizing the vegetation types in a table would not be suitable, as the detailed prose is critical for conveying the ecological and contextual nuances of each vegetation type.
We hope the explanation given in the initial response sufficiently clarifies our rationale, but please feel free to reach out if further clarification is needed.
- Line 155-217 Make a similar table to previous information table, please
Thank you for your comment. As this feedback overlaps with the previous suggestion (Line 157–171), we have already addressed it by consolidating the information into Table 1 for easier comparison and readability.
We hope this solution resolves your concerns, but please let us know if there are any additional adjustments needed.
- All those details of 25 species should be moved to supplement part because it is too long in the main part. Suggest to keep critical information in a table to compare.
The detailed descriptions of the 25 species are an integral part of the work itself, as they form the foundation of our study and are critical for understanding the results and conclusions. Moving this information to the supplementary section would detract from the core narrative and make it harder for readers to follow the main arguments of the manuscript.
While we appreciate the suggestion to use a table for comparison, the level of detail provided is necessary to capture the nuances of each species, which would not be adequately conveyed in a condensed tabular format. We believe that keeping these descriptions in the main text ensures clarity and maintains the scientific rigor of the study. Please let us know if there are specific adjustments you would recommend to enhance readability while preserving the content in the main section.
- Line 253,343,355,356,495 What are the meaning of question marks in those lines?
Corrected, we added the meaning of this abbreviation in the Results section. Thank you for pointing this out. The question marks indicate "Localities not confirmed" in taxonomy. This notation is used when describing the distribution of a species in cases where its presence in a certain locality has not been confirmed.
- Line 567-576 Suggest to make a table, please
Thank you for your suggestion. While we understand the intent behind organizing the information into a table for clarity, it would be challenging to adequately represent this level of detail and complexity in a tabular format. The prose in this section is critical for capturing the ecological and geographical associations of the species, which would be lost or oversimplified in a table.
The grouping and explanation provided here are integral to understanding the relationships between species, vegetation types, and distribution patterns, and we believe that the current narrative format conveys this information more effectively.
- The discussion part is quite similar to the result part, it doesn’t look like to discuss and has to be reorganized. Especially, the last paragraph is quite similar to previous result needs to be rewritten or deleted.
Corrected, we added a whole new part of the Discussion, better discussing the results we obtained, also following the suggestions from Reviewer 1.
- All the format of references are not correct because there is no information of issues. Too many really old references, suggest to find some new references.
The reference format is in line with the journals’ requirements, as the issue number is not required; we checked in the latest published articles in Insects, like those:
https://www.mdpi.com/2075-4450/15/12/960 ; https://www.mdpi.com/2075-4450/15/12/957 ; https://www.mdpi.com/2075-4450/15/12/954 (to make few examples), and our literature style matches the one of those.
Furthermore, it is quite obvious that the reported literature is rather old, as the description of the species belonging to the genus analyzed is referred to those (past) years. It would be a great taxonomic error if, otherwise, we would have not cited those.
Round 2
Reviewer 2 Report
Comments and Suggestions for Authors
It is good to see the improving of this manuscript. Still have two question need to be solved:
1. As a research paper, I still think it is too long and limit to read for effective information. Suggest to squeeze it a bit because I don't think some details are necessary.
2. It is not a suitable way to mention 'the updated geographical distribution and ecological information are reported (areas not confirmed are indicated with a question mark (?))' Suggest to delete those locations or expain the reason why they can not be comfirmed.
Author Response
1. As a research paper, I still think it is too long and limit to read for effective information. Suggest to squeeze it a bit because I don't think some details are necessary.
Thank you for your advice. However, this research contribution deals with a topic for which, apart from our recent works, the relevant bibliography is stuck in the 1950s or earlier. Therefore, we believe it is essential to provide a complete framework of the entire issue and none of the details reported are superfluous. We have carefully reread the manuscript and have eliminated a paragraph of 17 lines. We hope that this change will be satisfactory.
2. It is not a suitable way to mention 'the updated geographical distribution and ecological information are reported (areas not confirmed are indicated with a question mark (?))' Suggest to delete those locations or expain the reason why they can not be comfirmed.
Thank you for your comment. We have deleted the question marks (?) in the text.